# Early conversion to a CNI-free immunosuppression with SRL after renal transplantation—Long-term follow-up of a multicenter trial

Joachim Andrassy[1]*, Markus Guba[1], Antje Habicht[2], Michael Fischereder[3], Johann Pratschke[4], Andreas Pascher[4,5], Katharina M. Heller[6], Bernhard Banas[7], Oliver Hakenberg[8], Thomas Vogel[5], Bruno Meiser[2], Andrea Dick[9], Jens Werner[1], Teresa Kauke[1,9], for the SMART-Study Group¶

1 Department of General, Visceral, and Transplant Surgery, Ludwig-Maximilian's University, Campus Grosshadern, Munich, Germany, 2 Transplant Department, Ludwig-Maximilian's University, Campus Grosshadern, Munich, Germany, 3 Department of Medicine, MED IV, Nephrology, Ludwig-Maximilian's University, Campus Grosshadern, Munich, Germany, 4 Department of General, Visceral, and Transplant Surgery, Charité, Campus Virchow-Clinic, Berlin, Germany, 5 Department of General, Visceral, and Transplant Surgery, University of Münster, Münster, Germany, 6 Department of Medicine, Division of Nephrology, University of Erlangen, Germany, 7 Department of Internal Medicine II, Nephrology and Transplantation, University Medical Center, Regensburg, Germany, 8 Department of Urology, University of Rostock, Germany, 9 Laboratory for Immunogenetics, Division of Transfusion Medicine, Cell Therapeutics and Hemostaseology, Ludwig-Maximilian's University, Munich, Germany

¶ Membership of the SMART-Study Group is listed in the Acknowledgments.
* joachim.andrassy@med.uni-muenchen.de

**Data Availability Statement:** All relevant data are within the paper and its Supporting Information files.

## Abstract

### Introduction

Early conversion to a CNI-free immunosuppression with SRL was associated with an improved 1- and 3- yr renal function as compared with a CsA-based regimen in the SMART-Trial. Mixed results were reported on the occurrence of donor specific antibodies under mTOR-Is. Here, we present long-term results of the SMART-Trial.

### Methods and materials

N = 71 from 6 centers (n = 38 SRL and n = 33 CsA) of the original SMART-Trial (ITT n = 140) were enrolled in this observational, non-interventional extension study to collect retrospectively and prospectively follow-up data for the interval since baseline. Primary objective was the development of dnDSA. Blood samples were collected on average 8.7 years after transplantation.

### Results

Development of dnDSA was not different (SRL 5/38, 13.2% vs. CsA 9/33, 27.3%; P = 0.097). GFR remained improved under SRL with 64.37 ml/min/1.73m$^2$ vs. 53.19 ml/min/1.73m$^2$ (p = 0.044). Patient survival did not differ between groups at 10 years. There was a

**Funding:** J. A. received funding from an unrestricted medical grant by Pfizer Pharma GmbH to perform this trial. The funders had no role in study design, data collection and analysis, decision to publish, or preparation of the manuscript.

**Competing interests:** This trial was supported by a restricted grant of Pfizer GmbH. This does not alter our adherence to PLOS ONE policies on sharing data and materials. Pfizer had no function in data management, analysis, interpretation nor in preparation of this manuscript.

**Abbreviations:** CAN, chronic allograft nephropathy; CIT, cold ischemia time; CNI, Calcineurininhibitor; CsA, Ciclosporin A; DCGS, death censored graft survival; DGF, delayed graft function; DSA, donor specific antibody; eCrCl, estimated creatinine clearance; GFR, glomerular filtration rate; HLA, human leucocyte antigen; MFI, mean fluorescence intensity; OPTN, Organ Procurement and Transplantation Network; PRA, panel reactive antibody; SAB, single antigen beads; sCr, serum creatinine; SRL, Sirolimus; Tk, Time of conversion.

trend towards a reduced graft failure rate (11.6% SRL vs. 23.9% CsA, p = 0.064) and less tumors under SRL (2.6% SRL vs. 15.2% CsA, p = 0.09).

## Conclusions

An early conversion to SRL did not result in an increased incidence of dnDSA nor increased long-term risk for the recipient. Transplant function remains improved with benefits for the graft survival.

## Introduction

An mTOR-I based immunosuppression following renal transplantation remains ill accepted. Latest OPTN data indicate a continuous decline of mTOR-I use of currently below 5% [1]. Undisputed benefits for renal function and development of skin tumors [2, 3] are opposed by an array of side effects and limited tolerability [4, 5]. Furthermore, immunological potency when administered as main basic immunosuppressant was accused to lack behind Calcineurininhibitors (CNI) [5]. Despite all progress that has been made throughout the past decades graft survival remains limited. Ten year graft loss of deceased donor renal transplants has to be expected in 51.6% [1].

In trying to avoid the negative while preserving the positive effects of the mTOR-Is various conversion strategies had been studied [3, 6–8]. Pursuing a similar rationale many trials investigated the combination of mTOR-Is and CNIs [6, 9–11].

Irrespective of the trial design reliable "long-term" data are scarce. Follow-up universally stops well before the known half-lives of the grafts. And registry data which are preferably used to step in to close this gap of information are helpful but less accurate.

A growing body of evidence indicates that development of de novo donor specific antibodies (dnDSA) is a strong risk factor for the graft survival [12–14]. The incidence of dnDSAs is increasing over time and is thought to be around 20–30% after 5 years of transplantation with implications for the occurrence of humoral rejection [15]. The question if dnDSAs occur more often under mTOR-Is compared to CNIs remains unclear. Patients from the ZEUS and HERACLES trials showed a higher percentage of dnDSA after conversion to EVRL (23%) compared to the CsA (11%) [16]. This could not be confirmed by the TRANSFORM nor the PostConcept-Study [17, 18].

In 2006, we initiated the randomized controlled multicenter "SMART"-Trial where an early switch from CsA to SRL after renal transplantation was evaluated. One- and three-year results have been published [3, 19]. Here, we deliver "long-term" data (~ 10 years) of this trial on donor-specific antibodies, transplant function, graft and patient survival.

## Materials and methods

### Trial design

This trial was conducted according to GCP guidelines and the declaration of Helsinki and was approved by the local ethics committee of the Ludwig-Maximilian's university (LMU in Munich) and the ethics committees of the participating centers (EudraCT-Nr. 2013-004956-39).

This trial was a "follow-up" analysis of a prospective randomized controlled multicenter trial following renal transplantation (SMART-Trial; Controlled-trials.com, ISRCTN no.

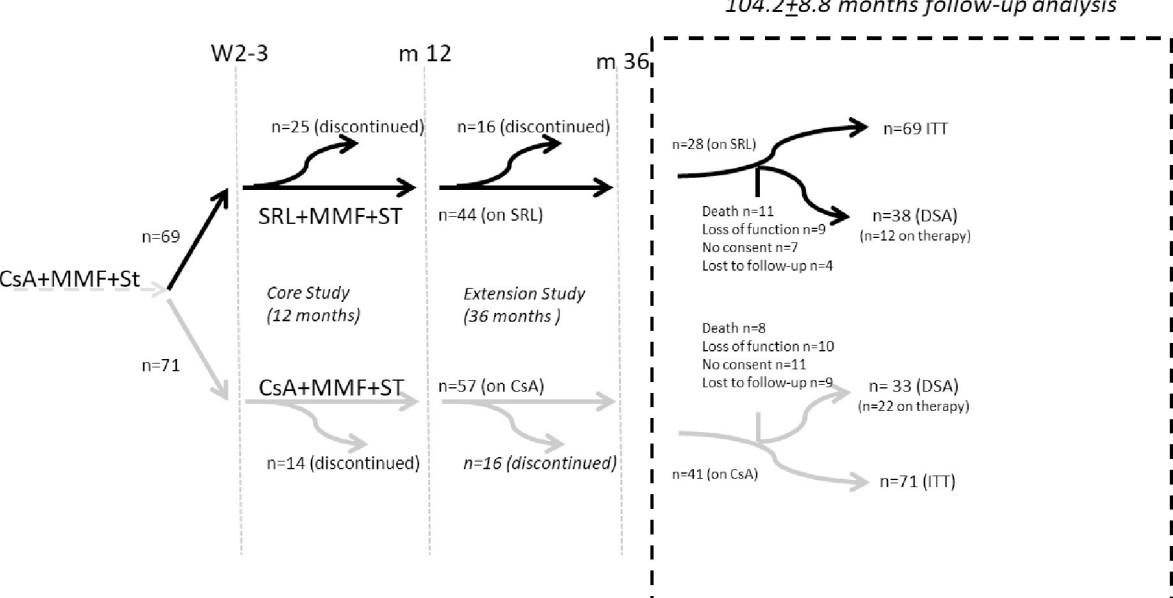

**Fig 1. Flowchart.** This is a follow-up trial of the randomized, controlled multicenter SMART-Trial (area shown with dashed lines). 12- and 36 months' data have already been published. The original ITT population consisted of n = 140 patients. Data on all patients were used for graft and patient survival analysis. n = 74 (SRL n = 39 and CsA n = 35) appeared to a follow up visit of which n = 71 (SRL n = 38 and CsA n = 33) had still a functioning graft and could thus be included for this follow-up trial for antibody screening and transplant function tests on average 104.2+8.8 months after the transplantation.

74429508). For the original trial none of the transplant donors were from a vulnerable population and all donors or next of kin provided written informed consent that was freely given.

The original design, 12- and 36-months data have been described in detail elsewhere [3, 19]. In short, n = 140 patients were started on a regular Cyclosporine A (CsA-) containing regimen and randomized to either remain on CsA or be switched to a CNI-free Sirolimus (SRL)-based immunosuppressive regimen between day 10–24 after transplantation (Fig 1). CsA was started in all patients within 24 hours after transplantation along with oral MMF. All patients received induction therapy with ATG (Fresenius) and 500 mg methylprednisolone intraoperatively and a maintenance dose thereafter according to the local practice of the participating center. SRL was initiated with a loading dose of up to 0.1 mg/kg followed by 2 to 4 mg/d once daily, aiming for an initial target trough level of 8 to 12 ng/mL. At this time, CsA was reduced by 50% and eliminated 3 days later. After 3 months, SRL was tapered to achieve target levels of 5 to 10 ng/ml.

## Eligibility criteria

All patients originally randomized to the SMART trial (ITT-cohort) were included and contacted by mail using a study plan including a consent form. In case patients did not respond they were contacted via telephone in the next step. All patients were > 18 years of age. One cohort consisted of n = 71 patients with still functioning graft who personally appeared to a control visit into the transplant centers. These patients delivered the blood samples for donor specific antibody- and current kidney function testing. Data on all SMART ITT-patients (n = 140) were gathered by retrospective chart review and contact of the primary care physician as well as the patients themselves. These data were used for the analysis of graft and patient survival.

## Primary objective

Incidence and characterization of donor specific antibodies.

## Secondary objectives

Patient and graft survival, transplant function, acute rejection episodes, incidence of malignancy and infection, therapy discontinuations, adverse events.

## HLA-antibody testing

All samples were sent to and processed in the Laboratory of Immunogenetics, Ludwig-Maximilian's-University in Munich, Germany.

Serum samples were screened for HLA antibodies using Luminex-technology with LABScreen Single Antigen Beads (SAB) (One Lambda, Canoga Park, CA, USA). The C1q-binding capacity of antibodies was also tested by C1q-SAB assay (One Lambda, Canoga Park, CA, USA). Donor specificity of HLA-antibodies was assumed for a MFI cut-off higher than 1000. All tests were performed according to the manufacturer's specifications.

## Statistics

Data are summarized by descriptive statistics based on mean and standard deviation for continuous parameters or absolute and relative frequencies for categorical variables. Comparisons between groups were performed by use of the nonparametric Wilcoxon rank sum test for continuous variables and Fisher's exact test for the analysis of contingency tables. All p-values were two-sided, and a $p < 0.05$ was considered statistically significant.

Patient and graft survival (and other time to event data) were calculated according to the Kaplan Meier method and compared between randomized groups using the log-rank test. Actuarial 5 and 10-year survival rates were calculated based on the Life Table Method. A cox model was applied for the estimation of hazard ratios. Transplant function was further assessed by comparing the changes from month 3 after the transplantation to the end of follow-up. For these, mean changes from baseline were analyzed using a maximum likelihood (ML)-based repeated measures approach. Analyses include the fixed, categorical effects of treatment, and the continuous time point as well as their interaction. A first-order autoregressive covariance structure was applied to model the within-patient errors. Occurrence of de novo HLA antibodies were compared between treatment arms by use of the site adjusted Mantel-Haenszel test. Univariate analysis of potential factors influencing the development of DSA was performed using Fisher's exact test with a threshold of $p < 0.1$. Continuous parameters were dichotomized based on cut-points evaluated by ROC analysis using the Youden index. Variables for analysis were selected when there was a frequency of at least 5.

Statistical analyses were done using SAS for windows, version 9.3 (SAS Institute Inc., Cary, NC, USA).

## Results

### Demographics

Of the n = 140 patients randomized to the original SMART-trial, n = 71 patients (n = 38 SRL; n = 33 CsA) presented for study examination on average 8.7 years after the transplantation (104±9.5 months SRL vs. 104±8.1 months CsA; p = 0.89). Age, height, weight, gender, ethnicity, underlying condition showed no significant differences between the groups (Table 1). The immunological risk as defined by the panel reactive antibodies (PRAs) pre-Tx was similar

**Table 1. Demographics.**

| | SMART Population | | | Long Term Follow Up (SMART-DSA) | | |
|---|---|---|---|---|---|---|
| | SRL (N = 69) N (%) or mean ± SD | CsA (N = 71) N (%) or mean ± SD | P- value | SRL (N = 38) N (%) or mean ± SD | CsA (N = 33) N (%) or mean ± SD | P-value |
| Recipient Age (yrs) | 47.0±10.8 | 47.1±11.1 | 0.9418 | 45.3±10.6 | 45.4±11.3 | 0.8310 |
| Height (cm) | 171.0±8.8 | 172.4±9.0 | 0.2769 | 171±7.8 | 171±8.5 | 0.6031 |
| Weight (kg) | 71.0±12.5 | 76.3±12.1 | 0.0158 | 69.5±12.2 | 74.2±11.0 | 0.0967 |
| Male | 45 (65.2) | 50 (70.4) | 0.5882 | 23 (60.5) | 23 (69.7) | 0.4636 |
| Polycystic Kidney Disease | 10 (14.5) | 8 (11.3) | 0.6205 | 4 (10.5) | 3 (9.1) | 1.0000 |
| Glomerulonephritis | 25 (36.2) | 30 (42.3) | 0.4930 | 13 (34) | 11 (33) | 1.0000 |
| PRA > 0 | 1 (1.4) | 2 (2.8) | 1.0000 | 0 (0.0) | 1 (3.0) | 0.4648 |
| CIT (hrs) | 12.1±5.7) | 13.0±7.0 | 0.5228 | 11.0±5.9 | 12.5±6.9 | 0.4122 |
| HLA-Mismatch | 2.8±1.2 | 2.9±1.2 | 0.6533 | 2.1±1.5 | 2.4±1.5 | 0.4445 |
| 1st Transplant | 62 (89.9) | 67 (94.4) | 0.3628 | 34 (89.5) | 32 (97.0) | 0.3633 |
| CMV-Status Donor +/rec- | 10 (14.5) | 10 (14.4) | 1.0000 | 6 (15.8) | 5 (15.2) | 1.0000 |
| DGF, Dialysis >1 | 15 (21.7) | 19 (26.8) | 0.5565 | 6 (15.8) | 9 (27.3) | 0.2600 |
| Donor Age (yrs) | 46.9±14.3 | 47.1±14.3 | 0.9451 | 46.5±13.6 | 45.3±14.0 | 0.6736 |
| Living Donation | 8 811.6) | 7 (9.9) | 0.7901 | 6 (15.8) | 2 (6.1) | 0.2705 |
| Ethnicity Caucasian | 68 (98.6) | 80 (98.6) | 1.0000 | 38 (100) | 33 (100) | 1.0000 |
| Therapy discontinuations | 46 (66.7) | 24 (33.8) | 0.0002 | 26 (68.4) | 11 (33.3) | 0.0043 |

Patients from both treatment arms did not differ significantly regarding underlying condition, immunization status pre transplant, DGF or specifics to the donor. There were significantly more therapy discontinuations in the SRL arm.

(0% > 0 SRL vs. 3.0% > 0 CsA, p = 0.46). Neither was there a difference with respect to HLA mismatch (2.1±1.5 SRL vs. 2.4±1.5 CsA, p = 0.44).

Altogether, n = 69 patients could not be included in the DSA-analysis (n = 31 SRL vs. n = 38 CsA, p = 0.31.; Fig 1) for the following reasons (Table 2): n = 19 had already died (n = 11 SRL vs. n = 8 CsA; p = 0.47), n = 13 patients were lost to follow-up (n = 4 SRL vs. n = 9 CsA; p = 0.24), n = 18 (n = 7 SRL vs. n = 11 CsA; p = 0.45) denied participation in this trial and n = 19 had lost transplant function (n = 9 SRL vs. n = 10 CsA; p = 1.0).

Upon presentation only n = 12 (31.6%) patients of the SRL arm and n = 22 (66.7%) of the CsA arm were still on the original immunosuppressant (p = 0.004, Table 1). Most patients had been switched to Tacrolimus in the meantime (S1 & S2 Tables).

**Table 2. Screening failures.**

| | SRL (N = 69) n (%) | CsA (N = 71) n (%) | P-value |
|---|---|---|---|
| Death | 11 (15.94) | 8 (11.27) | 0.4668 |
| Loss of function | 9 (13.04) | 10 (14.08) | 1.0000 |
| No consent | 7 (10.14) | 11 (15.49) | 0.4506 |
| Lost to follow-up | 4 (5.80) | 9 (12.68) | 0.2442 |
| All | 31 (44.93) | 38 (53.52) | 0.3171 |

There were no significant differences to screening failures in both treatment arms.

**Table 3. Analysis of de novo HLA antibodies.**

|  | SRL (N = 38) n (%) | CsA (N = 33) n (%) | P-value |
|---|---|---|---|
| Dn HLA-Ab | 9 (23.7) | 12 (36.4) | 0.1616 |
| Class I | 6 (15.8) | 4 (12.1) | 0.4772 |
| Class II | 6 (15.8) | 9 (27.3) | 0.2241 |
| C1q-binding | 6 (15.8) | 4 (12.1) | 0.6371 |
| DSA | 5 (13.2) | 9 (27.3) | 0.0968 |
| Class I | 2 (5.3) | 2 (6.1) | 0.8484 |
| Class II | 4 (10.5) | 8 (24.2) | 0.1198 |
| C1q-binding | 4 (10.5) | 3 (9.1) | 0.7274 |
| NDSA | 8 (21.1) | 6 (18.2) | 0.7553 |
| Class I | 6 (15.8) | 4 (12.1) | 0.4772 |
| Class II | 4 (10.5) | 3 (9.1) | 0.6253 |
| C1q-binding | 4 (10.5) | 3 (9.1) | 0.8932 |

No differences for the treatment groups could be detected regarding development of dn HLA-Abs.

## Development of de novo HLA antibodies

In n = 50 pts. (70%) no HLA-antibodies were found at the study visit (Table 3). In n = 21 pts. (30%) HLA-antibodies were positive (n = 9 (24%) SRL vs. n = 12 (36%) CsA; p = 0.16). C1q-binding ability could be confirmed in n = 10 of these HLA-antibody positive pts. (n = 6 (15.8%) SRL vs. n = 4 (12.1%) CsA; p = 0.64). HLA-antibodies directed against the donor specificity were found in n = 14 pts (20%) (n = 5 (13.2%) SRL vs. n = 9 (27.3%) CsA; p = 0.09). The majority of DSA was directed against HLA-class II antigens. In the non-donorspecific HLA-antibody positive patients we found even distribution of HLA-class I and II.

## Correlation of HLA-antibodies with transplant function and acute rejection

Renal function as measured by eGFR was not significantly different in pts. with DSA (DSA pos. 57.72±39.45 ml/min vs. DSA neg. 59.70±18.88 ml/min; p = 0.12) (Table 4). Renal function was significantly reduced in the presence of antibodies against HLA-class II (HLA II Abs: 46.18±17.22 ml/min vs. no HLA II Abs: 62.82±24.42 ml/min; p = 0.01).

## Risk for de novo DSA development

Univariate analysis identified only the recipient age < 39 years (OR: 3.07; 0.92–10.29; p = 0.09; Table 5) as a risk factor for de novo DSA development. Male gender (OR: 4.06; 0.83–19.86;

**Table 4. Correlation of DSA and Class II HLA antibodies with transplant function.**

| Renal Function | mean±SD | mean±SD | P-value |
|---|---|---|---|
| DSA | Yes (n = 14) | No (n = 57) |  |
| eGFR (Nankivell, mL/min/1.73 m2) | 57.72±39.45 | 59.70±18.88 | 0.1203 |
| sCr (mg/dl) | 2.02±0.98 | 1.59±0.69 | 0.0564 |
| Class II | Yes (N = 15) | No (N = 56) |  |
| eGFR (Nankivell, mL/min/1.73 m2) | 46.18±17.22 | 62.82±24.42 | 0.0101 |
| SCr (mg/dl) | 2.17±0.95 | 1.55±0.65 | 0.0063 |

Transplant function was significantly impaired when DSA or class II HLA Abs were found.

**Table 5. Univariate analyses on the risk for developing *de novo* DSA.**

|  | Univariate analysis | | |
| --- | --- | --- | --- |
|  | **Odds Ratio** | **95% CI** | **P** |
| Male | 4.06 | 0.83–19.86 | 0.1163 |
| Re-transplantation | 3.00 | 0.45–19.97 | 0.2537 |
| Rec. Age ≤ 39 | 3.07 | 0.92–10.29 | 0.0995 |
| Living donor | 2.84 | 0.59–13.66 | 0.1864 |
| CIT > 11h | 0.43 | 0.13–1.46 | 0.2351 |
| Low ATG induction | 2.84 | 0.59–13.66 | 0.1864 |
| Donor age ≤ 57 | 4.23 | 0.51–35.31 | 0.2731 |
| *SCr-Tk+7 ≥ 1.27 | 5.07 | 0.61–42.03 | 0.1625 |
| Banff 4 | 1.76 | 0.53–5.87 | 0.3587 |
| Ciclosporin | 2.47 | 0.74–8.33 | 0.2311 |

* Serum Creatinine 7 days after the timepoint of conversion

p = 0.12), living donation (OR: 2.84; 0.59–13.66; p = 0.19), low ATG induction (OR: 2.84; 0.59–13.66; p = 0.19) and an impaired transplant function (OR: 5.07; 0.61–42.03; p = 0.16) were not significant.

## Transplant function

Transplant function improved under SRL starting with the randomization and remained improved until the latest measurement 104±9 months after the transplantation (Fig 2; Table 6; SRL 64.37±26.44 ml/min/1.73 m$^2$ vs. CsA 53.19±19.83 ml/min/1.73 m$^2$; p = 0.04). Measurements by Cockcroft-Gault (SRL 56.03 ± 18.62 ml/min/1.73 m$^2$ vs. CsA 48.98 ± 19.93 ml/min/1.73 m$^2$; p = 0.12), MDRD (SRL 53.42 ± 21.28 ml/min/1.73 m2 vs. CsA 45.92 ± 20.87 ml/min/1.73 m2; p = 0.11) and CKD-EPI (SRL 53.86 ± 21.64 ml/min/1.73 m2 vs. CsA 45.78 ± 20.84

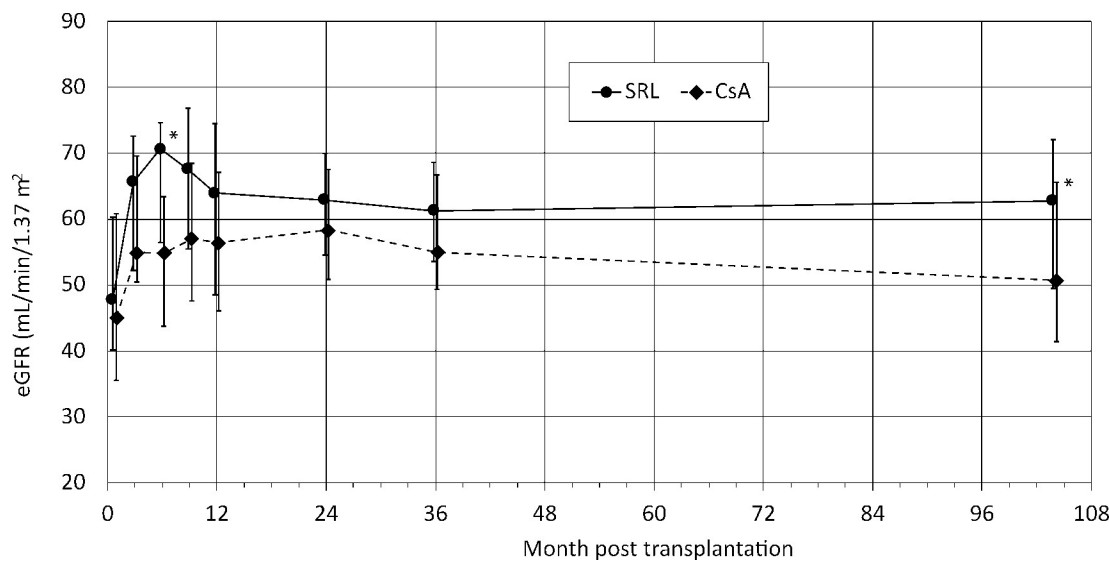

**Fig 2. Transplant function over time.** Transplant function was significantly better in the SRL treatment group at long term follow-up. Data shown are median values and interquartile ranges starting from randomization in patients who completed the DSA follow up at a median of 104 ± 9 months after transplantation. Significant p-values for the Wilcoxon rank sum test are marked with an asterisk.

**Table 6. Transplant function at long term follow up (104± 8.8 months after Tx).**

| | SRL | CsA | p-Value |
|---|---|---|---|
| **ITT population** | | | |
| sCr (mg/dL)) | (n = 38) | (n = 33) | |
| Mean ± SD | 1.54 ± 0.71 | 1.83 ± 0.81 | 0.0720 |
| eGFR (Nankivell, mL/min/1.73m$^2$) | (n = 38) | (n = 32) | |
| Mean ± SD | 64.37 ± 26.44 | 53.19 ± 19.83 | 0.0444 |
| eCrCl (Cockroft Gault, mL/min) | (n = 38) | (N = 32) | |
| Mean ± SD | 56.03 ± 18.62 | 48.98 ± 19.93 | 0.1211 |
| eGFR (MDRD, mL/ mL/min/1.73m$^2$) | (n = 38) | (n = 33) | |
| Mean ± SD | 53.42 ± 21.28 | 45.92 ± 20.87 | 0.1053 |
| eGFR (CKD-EPI, mL/ mL/min/1.73m2) | (n = 38) | (n = 33) | |
| Mean±SD | 53.86±21.64 | 45.78±20.84 | 0.1053 |
| **On therapy population** | | | |
| sCr (mg/dL)) | (n = 12) | (n = 22) | |
| Mean ± SD | 1.39 ± 0.49 | 1.74 ± 0.63 | 0.0937 |
| eGFR (Nankivell, mL/min/1.73m$^2$) | (n = 12) | (n = 21) | |
| Mean ± SD | 66.00 ± 15.25 | 52.83 ± 19.71 | 0.0314 |
| eCrCl (Cockroft Gault, mL/min) | (n = 12) | (n = 21) | |
| Mean ± SD | 57.05 ± 16.00 | 47.71 ± 19.58 | 0.1117 |
| eGFR (MDRD, mL/ mL/min/1.73m$^2$) | (n = 12) | (n = 22) | |
| Mean ± SD | 55.33 ± 17.74 | 45.34 ± 20.43 | 0.0869 |
| eGFR (CKD-EPI, mL/ mL/min/1.73m2) | (n = 12) | (n = 22) | |
| Mean±SD | 55.99±18.68 | 44.84±19.57 | 0.0869 |

Transplant function as measured by Nankivell was significantly improved for the SRL treatment group. Patients who had remained on SRL also showed a significant benefit compared to the CsA treatment.

ml/min/1.73 m2; p = 0.11) missed significance. Analysis of those patients who had remained on the original therapy showed a similar picture with an improved transplant function under SRL.

GFR comparison of month 3 after Tx to most recently (104±9 months) revealed a more pronounced deterioration in the CsA group (MDRD: -0.87 ± 14.58 ml/min/1.73 m$^2$ SRL vs. -8.26 ± 18.04 ml/min/1.73 m$^2$ CsA; p = 0.07; CKD-EPI: -2.08 ± 15.39 ml/min/1.73 m$^2$ SRL vs. -9.91 ± 18.59 ml/min/1.73 m$^2$ CsA; p = 0.06; Table 7).

Mixed model longitudinal analysis of renal function with fixed effects of randomized treatment, time and the combination of time and treatment confirmed a significant advantage of the SRL group starting at 3 months after transplantation (S3 Table).

## Patient survival

Looking at the original ITT cohort of n = 140 patients, Kaplan-Meier curves did not show a difference for the patient survival (Fig 3; p = 0.67; HR 1.225 (95% CI: 0.483–3.104)). Actuarial five-year survival was on average 94.2% (SRL: 95.5% vs. CsA 92.9%) and 82.8% after ten years (SRL: 83.6% vs. CsA 82.1%). Under SRL n = 11 patients (16%) died compared to n = 8 (11%) in the CsA arm (p = 0.47).

Causes of death were: n = 3 cardiovascular (2 SRL vs. 1 CsA), n = 3 malignancy (2 SRL vs. 1 CsA), n = 4 infectious (1 SRL vs. 3 CsA), n = 7 unknown (5 SRL vs. 2 CsA); 1 accident (CsA), n = 1 pulmonary embolism (SRL).

**Table 7. Change in eGFR from month 3 to 104±8.8 months post transplantation.**

| | SRL | CsA | p-Value |
|---|---|---|---|
| **ITT population** | | | |
| Δ-sCr (mg/dL)) | (n = 38) | (n = 33) | |
| Mean ± SD | -0.01 ± 0.57 | 0.27 ± 0.68 | 0.1154 |
| Δ-eGFR (Nankivell, mL/min/1.73m$^2$) | (n = 38) | (n = 32) | |
| Mean ± SD | 0.17 ± 14.31 | -6.46 ± 18.12 | 0.1733 |
| Δ-eCrCl (Cockroft Gault, mL/min) | (n = 38) | (n = 32) | |
| Mean ± SD | -3.61 ± 14.17 | -11.01 ± 18.77 | 0.0760 |
| Δ-eGFR (MDRD, mL/ mL/min/1.73m$^2$) | (n = 38) | (n = 33) | |
| Mean ± SD | -0.87 ± 14.58 | -8.26 ± 18.04 | 0.0677 |
| Δ-eGFR (CKD-EPI, mL/ mL/min/1.73m2) | (n = 38) | (n = 33) | |
| Mean±SD | -2.08±15.39 | -9.91±18.59 | 0.0643 |
| **On therapy population** | | | |
| Δ-sCr (mg/dL)) | (n = 12) | (n = 22) | |
| Mean ± SD | -0.12 ± 0.60 | 0.22 ± 0.51 | 0.2269 |
| Δ-eGFR (Nankivell, mL/min/1.73m$^2$) | (n = 12) | (n = 21) | |
| Mean ± SD | 3.33 ± 14.38 | -7.26 ± 20.13 | 0.2385 |
| Δ-eCrCl (Cockroft Gault, mL/min) | (n = 12) | (n = 21) | |
| Mean ± SD | -2.20 ± 14.46 | -12.23 ± 20.51 | 0.1393 |
| Δ-eGFR (MDRD, mL/ mL/min/1.73m$^2$) | (n = 12) | (n = 22) | |
| Mean ± SD | 1.22 ± 15.66 | -9.29 ± 19.64 | 0.1653 |
| Δ-eGFR (CKD-EPI, mL/ mL/min/1.73m2) | (n = 12) | (n = 22) | |
| Mean±SD | -0.26±16.37 | -11.18±20.08 | 0.2318 |

For patients from the CsA treatment group all measurements showed a deterioration of the transplant function over this observation period. Under SRL, transplant function remained more stable with either no or minimal change of function compared to month 3. ΔsCr: delta serum creatinine, ΔeCrCl: delta estimated creatinine clearance, ΔeGFR: delta estimated glomerular filtration rate (Differences: follow up month 3).

## Graft survival

Graft survival was not significantly different between treatment arms. Actuarial five-year graft survival was 87.6% (SRL: 89.6% vs. CsA: 85.7%) and ten-year graft survival was 60.2% (SRL: 68.8% vs. CsA: 52.0%). There was a trend towards a reduced graft failure rate under SRL (11.6% SRL vs. 23.9% CsA). Beginning at 8–9 years after the transplantation, Kaplan-Meier curves show a particularly increased death censored failure rate for the CsA treated patients (p = 0.064, Fig 4). The median was not yet reached in both treatment arms. Graphical and numerical methods were applied for checking the adequacy of the Cox regression model and the decision finally based on the Kolmogorov-type supremum test based on 1,000 simulations. With P = 0.1040 the assumption of proportional hazards can be accepted.

A Cox proportional hazard model revealed a 0.461 times smaller hazard in the SRL group compared to CsA (S4 Table; 95% CI; 0.199–1.069). There was no relevant difference for the actuarial DCGS five years after the transplantation (SRL: 93.9% vs. CsA 90.9%). This changed for the ten-year analysis where the benefit under SRL almost reached statistical significance (SRL 81.8% vs. CsA 56.4%).

## Adverse events

The adverse events were recorded for the n = 71 patients who had appeared for a control visit and delivered a blood sample. Proteinuria was recorded for n = 10 patients, n = 3 for SRL and

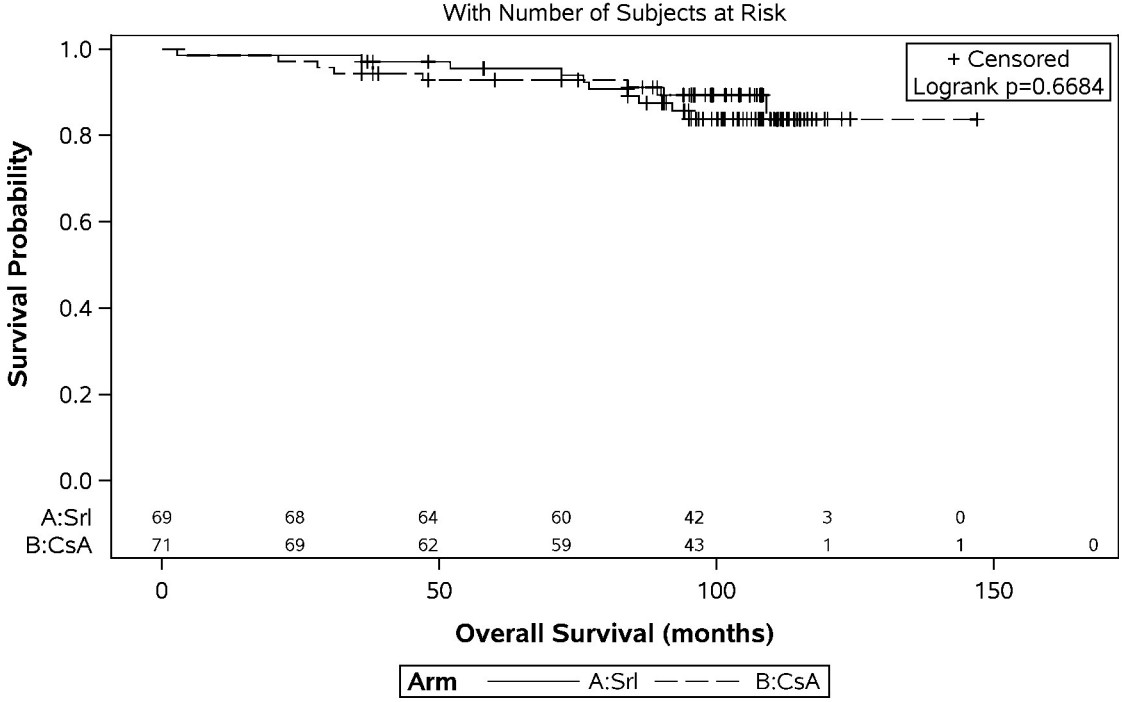

**Fig 3. Kaplan-Meier curve on patient survival.** Event rates were 10/69 in the SRL and 8/71 in the CsA Group. Hazard ratio for SRL (95%CI): 1.225 (0.483–3.104).

n = 7 for CsA (p = 0.17). There was no difference for combined biopsy proven and suspected acute rejections between the two treatment arms (0% SRL vs. 8.6% CsA; p = 0.1; Table 8). There was no significant difference for infections, cardiovascular events or metabolic disorders. Malignancy occurred in n = 1 (2.6%) under SRL and in n = 5 (15.2%) under CsA, (p = 0.09). For those patients remaining on therapy (SRL n = 12 vs. CsA = 22) there was no malignancy recorded under SRL vs. n = 5 under CsA (p = 0.06). With respect to the other adverse events no significant further findings could be reported, likely due to the low numbers.

## Discussion

Many trials exist comparing the effects of an early switch to mTOR-Is with a CNI-based immunosuppression [18–26]. Irrespective of the type of the mTOR-I, most of these trials confirm an improved renal function especially in those patients who remain "on therapy" [20, 21] and an efficacy rate in terms of acute rejection, graft and patient survival which is similar to that of CNIs. Besides, the rate of certain malignant and viremic diseases is reduced [27, 28]. However, not all trials could report a favorable outcome especially in earlier times, when higher trough levels and loading doses were used and the experience with side effects was limited [5, 29–31].

One challenge of the current times is the acquisition of reliable "long-term" data reaching beyond the reported half-lives of the grafts. The question for example, if the improved transplant function and the antiproliferative effect with less CAN [32, 33] under mTOR-Is will ultimately translate into a prolonged transplant survival remains unclear to this date.

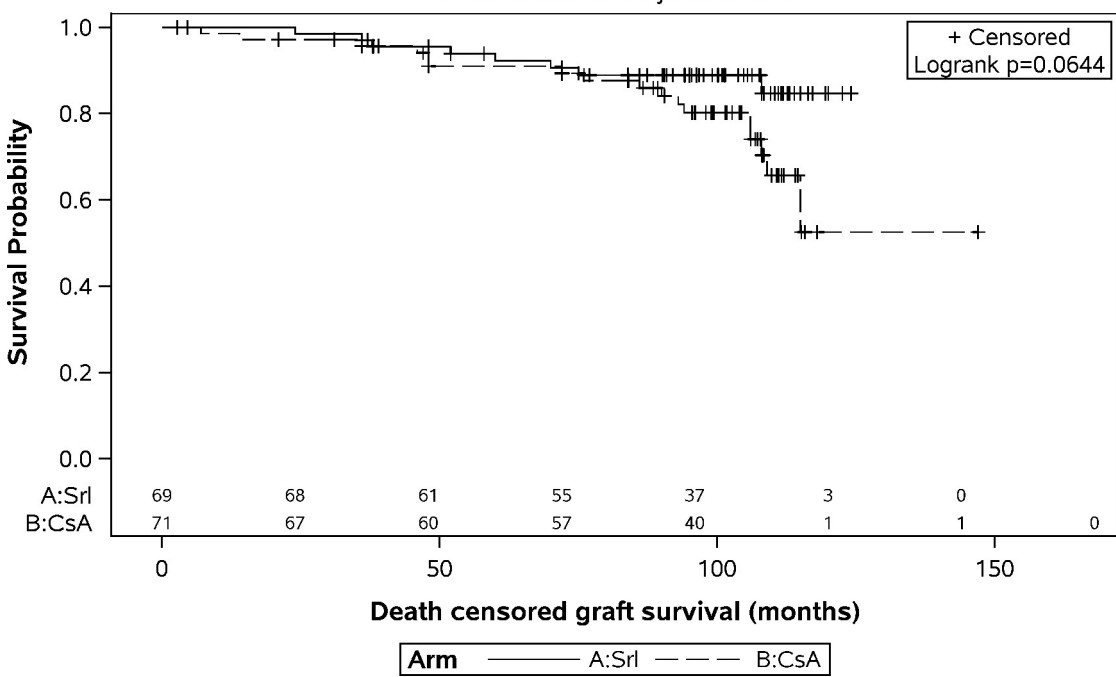

**Fig 4. Kaplan-Meier curve on death censored graft survival.** Failure rates were 8/69 in the SRl and 17/71 in the CsA Group. Hazard ratio for SRL (95%CI): 0.461 (0.199–1.069).

Here, we present follow-up data on a randomized controlled multicenter trial on renal transplant patients receiving SRL after a short course of CsA (up until 21 days). Of the original ITT cohort (n = 140 randomized patients), n = 71 patients with functioning grafts delivered blood samples with information on dnDSA and transplant function 8.7 years on average after the transplantation.

Primary focus was the analysis of donor specific HLA antibodies. Humoral immunity plays a dominant part for deterioration of graft function and graft loss [12, 34]. Preexisting DSA predispose for early antibody-mediated rejection and poorer graft survival [15]. Following renal transplantation DSAs have been shown to develop de novo in 13–27% of previously non-sensitized patients, mostly within the first year but also many years thereafter [35, 36]. De novo DSA are also associated with antibody-mediated rejection (AMR), chronic allograft dysfunction and diminished allograft survival [13, 37, 38]. In particular, antibodies binding the

**Table 8. Adverse events.**

|                        | SRL N = 38 (%) | CsA N = 33 (%) | P-value |
|------------------------|----------------|----------------|---------|
| Proteinuria            | 3 (7.9)        | 7 (21.2)       | 0.1712  |
| Malignancy             | 1 (2.63)       | 5 (15.15)      | 0.0900  |
| Acute Rejections       | 0 (0.00)       | 3 (9.09)       | 0.0955  |
| Infections             | 9 (23.68)      | 7 (21.21)      | 1.0000  |
| Cardiovascular events  | 8 (21.05)      | 3 (9.09)       | 0.2022  |
| Metabolic disorders    | 5 (13.16)      | 4 (12.12)      | 1.0000  |

Adverse events are reported here only for the extended follow up after the M36 visit.

complement fraction C1q, which is the first step in the activation of the classic complement cascade, have been shown to be a risk factor for the development of AMR [39].

In this trial we screened the sera of n = 71 patients for the presence of HLA-antibodies by means of SAB and additionally we tested the C1q-binding capacity of HLA-antibodies. We observed that n = 21 (30%) patients had HLA-antibodies and C1q-binding capacity could be confirmed in n = 10. The incidence of non-complement-binding and complement-binding HLA-antibodies in our study population was within the range of previously published reports [40, 41]. In accordance with our previous report we observed no statistically significant difference between C1q-binding and C1q-non-binding HLA-antibodies [40]. There was also no statistical difference in HLA-antibody positivity in the SRL and the CsA group (SRL 24% vs. CsA 36%, p = 0.16). De novo DSA were found in n = 14 (20%). Numerically, there were less dnDSA positive patients under SRL (5/38, 13.2%) compared to CsA (9/33, 27.3%) closely missing significance (p = 0.09). The results are mixed to this in the literature. Some trials reported a higher incidence of dnDSA under mTOR-Is [16, 42–45] and some did not [18, 46, 47]. The largest multicenter study to date comparing mTOR-Is with CNIs following renal transplantation (TRANSFORM) showed no negative effect for the mTOR-I in terms of dnDSA incidence and antibody-mediated rejection at 12 months [18]. When citing these trials, one has to be aware that a more sophisticated HLA-matching on the epitope level (eplet) would probably add further information to the choice of the maintenance immunosuppression in this regard.

We found a significant correlation between dnDSA positivity on impaired graft function when antibodies were directed against HLA-class II antigens (p = 0.01). This effect is in line with previously published data [48].

Results showed no significant difference in patient and graft survival. The latter, however, deserves further consideration. The actuarial 5-yr DCGS showed no difference between the two groups (SRL: 93.9% vs. CsA 90.9%). Kaplan-Meier curve of the DCGS shows a deterioration of the CNI- but not the mTOR-I-treated grafts beginning at around month 90–100 as one would expect according to the known half-lives of ~9–10 years. And the actuarial 10-yr- DCGS shows a trend towards a better survival under SRL (81.8% SRL vs. 56.4% CsA) averaging 68.3%. It is difficult to find reliable long-term data for a comparison. Our data correspond well to the latest OPTN/SRTR report. Here, the 10-yr graft survival was 48.4% (compared to 68.8% SRL and 52.0% CsA in our trial) [1]. Unfortunately, the report did not distinguish between the different immunosuppressants. For a better comparison this would have been helpful because only a minority had received a CsA- (1.7%) or an mTOR-I- (1.9%) based immunosuppression [1]. A single center trial with a follow-up of 7 yrs reported a substantially better actuarial 10-year graft survival of 63.5% and a DCGS of 77.5% for a CsA/MMF combination even without steroids [33]. But yet again, these results seem difficult to compare with because there were substantial differences in trial design, induction therapy and the percentage of living donation (71.5% vs. 5.7%).

Transplant function was shown to be superior in the SMART-trial under SRL 12 and 36 months after Tx [3, 19]. This could once again be confirmed in the current analysis at 104±8.8 months. On average, renal function had remained excellent in the SRL arm with an eGFR of 64.37±26.44 ml/min/1.73 $m^2$ which had even slightly improved compared to the measurement of month 3 after the transplantation. For those patients who had remained "on therapy" GFR was even better with 66.00 ± 15.25 ml/min/1.73 $m^2$. This is in accord with other publications which report that patients seem to benefit in particular when they remain "on therapy" [21, 49, 50]. In contrast, GFR had significantly deteriorated under CsA. Although difficult to compare for existing differences regarding the mTOR-I used, medication plan and study duration, these results appear similar to other trials with "longer" follow-ups, such as the ZEUS- or HERAKLES- 5-yr extension trials [21, 49]. Interestingly, decrease of transplant function under CNIs seems to occur well beyond the first 5 years [33, 50].

Benefits of an mTOR-I therapy regarding malignancy have been uniformly confirmed and recently shown to extend beyond skin cancers [27]. Thus, the results from this trial could be expected.

A limitation for this study is that 49% of the original study population could not be included for the analysis of DSAs. As outlined in the results section, most of these patients had either died, lost their graft or declined participation. Nonetheless, most of the relevant clinical information on these patients could still be gathered by retrospective chart review and contact of their primary care physicians and the patients themselves. With n = 4 in the SRL arm und n = 9 of the CsA the number of those who were actually "lost to follow-up" was much lower and appear acceptable considering the long follow-up.

Lack of tolerability remains an important aspect for the use of mTOR-Is and is another limitation of this trial. Only 31.6% (12/38) once randomized to receive SRL compared to 66.7% (22/33) started on CsA were still on the original immunosuppressant. The majority had stopped the SRL relatively early as only 40.6% had remained "on therapy" after the first 3 years. The first patients had been randomized by 2006 when the experience with mTOR-I side effects was low. This is an important aspect since many of these drug discontinuations would not have been pursued nowadays. Nonetheless, problems with the mTOR-I tolerability remain up to this day [18] and clinical experience is needed for a successful management. Patients in both treatment arms were usually switched to Tacrolimus. Even though the percentage of patients switched from SRL to TAC was higher compared to those switched from CsA, the overall exposure time to TAC was not significantly different between the groups. Another limitation specifically concerning our data on HLA-antibodies is that only a single measurement of HLA-antibodies in the post-transplant follow-up was used. Nonetheless, our data correlated well to the existing evidence. Lastly, we do not have data from histopathology to corroborate our findings.

In conclusion, we could show no difference for the occurrence of DSAs under SRL compared to CsA. The data confirmed an impaired graft function in the presence of DSAs. Graft function had remained significantly better under SRL vs. CsA with stable eGFR only under SRL compared to month 3 after the transplantation. Due to the long follow-up we could observe the expected gradual decline in graft survival under CsA and unexpectedly saw a benefit for the SRL therapy which did not become apparent until late in the observation period (8–9 yrs after Tx). Further trials with reliable long-term data will have to confirm our findings.

## Supporting information

**S1 Checklist. CONSORT 2010 checklist of information to include when reporting a randomised trial**\*.
(DOCX)

**S1 File.**
(CSV)

**S2 File.**
(CSV)

**S3 File.**
(CSV)

**S4 File.**
(CSV)

**S5 File.**
(CSV)

**S6 File.**
(CSV)

**S7 File.**
(CSV)

**S8 File.**
(CSV)

**S9 File.**
(CSV)

**S1 Protocol.**
(DOC)

**S1 Table. Therapy discontinuations and changes to Tacrolimus.**
(DOCX)

**S2 Table. Reasons for therapy changes to Tacrolimus.**
(DOCX)

**S3 Table.** A. Mixed model analysis of eGFR (Nankivell). B. Mixed model analysis of eGFR (MDRD).
(DOCX)

**S4 Table. Cox model for patient and death censored graft survival.**
(DOCX)

## Acknowledgments

Karl Fehnle (ALGORA Muenchen, Germany; monitoring of the study, data management and statistical analyses), Michael Eder (Department of Surgery, Munich University Hospital, Campus Grosshadern, Munich, Germany; monitoring of the study, data management).

## Author Contributions

**Conceptualization:** Joachim Andrassy, Teresa Kauke.

**Data curation:** Joachim Andrassy, Markus Guba, Antje Habicht, Michael Fischereder, Johann Pratschke, Andreas Pascher, Katharina M. Heller, Bernhard Banas, Oliver Hakenberg, Thomas Vogel, Bruno Meiser, Andrea Dick, Jens Werner, Teresa Kauke.

**Formal analysis:** Joachim Andrassy, Andrea Dick, Teresa Kauke.

**Funding acquisition:** Joachim Andrassy.

**Investigation:** Joachim Andrassy, Markus Guba, Antje Habicht, Michael Fischereder, Johann Pratschke, Andreas Pascher, Katharina M. Heller, Bernhard Banas, Oliver Hakenberg, Thomas Vogel, Bruno Meiser, Andrea Dick, Jens Werner, Teresa Kauke.

**Methodology:** Joachim Andrassy, Andrea Dick, Teresa Kauke.

**Project administration:** Joachim Andrassy, Teresa Kauke.

**Resources:** Joachim Andrassy, Michael Fischereder, Johann Pratschke, Andreas Pascher, Katharina M. Heller, Bernhard Banas, Oliver Hakenberg, Bruno Meiser, Jens Werner, Teresa Kauke.

**Supervision:** Joachim Andrassy, Teresa Kauke.

**Validation:** Joachim Andrassy, Markus Guba, Antje Habicht, Michael Fischereder, Johann Pratschke, Andreas Pascher, Katharina M. Heller, Bernhard Banas, Oliver Hakenberg, Thomas Vogel, Bruno Meiser, Andrea Dick, Jens Werner, Teresa Kauke.

**Visualization:** Joachim Andrassy, Teresa Kauke.

**Writing – original draft:** Joachim Andrassy, Teresa Kauke.

**Writing – review & editing:** Joachim Andrassy, Markus Guba, Antje Habicht, Michael Fischereder, Johann Pratschke, Andreas Pascher, Katharina M. Heller, Bernhard Banas, Oliver Hakenberg, Thomas Vogel, Bruno Meiser, Andrea Dick, Jens Werner, Teresa Kauke.

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
