## [Decision Letter · Decision Letter 0]

10 Dec 2019

PONE-D-19-25265

Early conversion to a CNI-free immunosuppression with SRL after renal Transplantation – long-term follow-up of a multicenter trial

PLOS ONE

Dear Prof. Dr. Andrassy,

Thank you for submitting your manuscript to PLOS ONE. After careful consideration, we feel that it has merit but does not fully meet PLOS ONE’s publication criteria as it currently stands. Therefore, we invite you to submit a revised version of the manuscript that addresses the points raised during the review process.

ACADEMIC EDITOR: 

The manuscript is of potential interest to the renal transplant community. However, it is not acceptable for publication in its current form. The major limitation of the study is a substantial drop-out rate and considerable tacrolimus exposure in both groups. Therefore additional data should be provided and discussion expanded:

The Authors should evaluate and compare the mean time with tacrolimus exposure in SRL vs CsA arm. Why were so many patients switched to tacrolimus?  A small table listing the reasons would be informative. Just listing late adverse events is not as informative as listing those that caused a change in drug delivery.Data on proteinuria in both groups should be provided.The Authors should highlight the limitations of the very low rate of patients on the original immunosuppression at the end of 8-9 years follow-up.The conclusions about dnDSA must consider that the majority of the patients were in use of tacrolimus that was not the original immunosuppression.The absence of histology of the grafts also limits the ability to make firm conclusions on the impact of CNI free status on graft stability, this limitation should be provided as well.

Additionally, please provide corrections according to the statistical Reviewer suggestions (Reviewer 6).

We would appreciate receiving your revised manuscript by Jan 23 2020 11:59PM. To enhance the reproducibility of your results, we recommend that if applicable you deposit your laboratory protocols in protocols.io, where a protocol can be assigned its own identifier (DOI) such that it can be cited independently in the future. For instructions see: http://journals.plos.org/plosone/s/submission-guidelines#loc-laboratory-protocols

We look forward to receiving your revised manuscript.

Kind regards,

Justyna Gołębiewska

Academic Editor

PLOS ONE

Journal Requirements:

**When submitting your revision, we need you to address these additional requirements:**

**Please ensure that your manuscript meets PLOS ONE's style requirements, including those for file naming. The PLOS ONE style templates can be found at http://www.plosone.org/attachments/PLOSOne_formatting_sample_main_body.pdf and http://www.plosone.org/attachments/PLOSOne_formatting_sample_title_authors_affiliations.pdf**Please provide additional details regarding participant consent. In the ethics statement in the Methods and online submission information, please ensure that you have specified (1) whether consent was suitably informed and (2) what type you obtained (for instance, written or verbal). If your study included minors under age 18, state whether you obtained consent from parents or guardians. If the need for consent was waived by the ethics committee, please include this information.In your Methods section, please provide additional information regarding the donated tissues or organs used in the study. Please specify whether the study involved the use of donated tissue/organs from any vulnerable populations, and provide information on the consent given by the donor or their next of kin. Examples of vulnerable populations include prisoners, subjects with reduced mental capacity due to illness or age, and children. If such a population as used, please ensure you have describe the population and justify the decision to use tissue/organ donations from this group. If not, please state in your Ethics Statement, 'None of the transplant donors were from a vulnerable population and all donors or next of kin provided written informed consent that was freely given.'Thank you for including your ethics statement:  "This trial was conducted according to GCP guidelines and the declaration of Helsinki and was approved by the local ethics committees of the participating centers."

a.Please amend your current ethics statement to include the full name of the ethics committee/institutional review board(s) that approved your specific study.

b.Once you have amended this/these statement(s) in the Methods section of the manuscript, please add the same text to the “Ethics Statement” field of the submission form (via “Edit Submission”).

5. Please amend the manuscript submission data (via Edit Submission) to include author Andreas Pascher.

6.Thank you for stating the following in the Financial Disclosure section:

J. A. received funding from an unrestricted medical grant by Pfizer Pharma GmbH to perform this trial.

We note that you received funding from a commercial source: Pfizer

7. 

We note that you have indicated that data from this study are available upon request. PLOS only allows data to be available upon request if there are legal or ethical restrictions on sharing data publicly. For information on unacceptable data access restrictions, please see http://journals.plos.org/plosone/s/data-availability#loc-unacceptable-data-access-restrictions.

Reviewers' comments:

Reviewer's Responses to Questions

**Comments to the Author**

1. Is the manuscript technically sound, and do the data support the conclusions?

Reviewer #1: Partly

Reviewer #2: Yes

Reviewer #3: Partly

Reviewer #4: No

Reviewer #5: Partly

Reviewer #6: Yes

2. Has the statistical analysis been performed appropriately and rigorously? 

Reviewer #1: No

Reviewer #2: Yes

Reviewer #3: Yes

Reviewer #4: Yes

Reviewer #5: Yes

Reviewer #6: Yes

3. Have the authors made all data underlying the findings in their manuscript fully available?

Reviewer #1: Yes

Reviewer #2: Yes

Reviewer #3: No

Reviewer #4: Yes

Reviewer #5: No

Reviewer #6: Yes

4. Is the manuscript presented in an intelligible fashion and written in standard English?

Reviewer #1: Yes

Reviewer #2: Yes

Reviewer #3: No

Reviewer #4: Yes

Reviewer #5: Yes

Reviewer #6: Yes

5. Review Comments to the Author

Reviewer #1: This is long-term data about the SMART study. This study aims to evaluate the eGFR in patients early converted to CNI-free immunosuppression with SRL after renal Transplantation. The authors discussed that were few studies with long term outcome about CNI-free immunosuppression.

This is a well writing manuscript with an extended follow-up of the primary study. The extension aims to evaluate the development of de novo donor-specific antibodies (dnDSA). This is a relevant question because the immunosuppression without CNI was associated with higher rates of dnDSA. A serious limitation was the sample size with very low patients in use of the original immunosuppression (31.6% in the sirolimus and 66.7% in the ciclosporin)

Main Questions:

1. Possible the good results of early convesion to sirolimus was related to the induction therapy with thymoglobulin. This type of induction therapy diverges from other CNI-convertions trials. The authors should highlight this point. Additionally, the use of a loading dose of sirolimus is not common practice today.

2. Measures of dnDSA. The authors show that there were no differences in dnDSA between the ciclosporin and the sirolimus group. However, this could be analysed with caution because only 31.6% of patients in the sirolimus group were in use of the original immunosuppression. The majority of the patients were switched to tacrolimus. The authors should evaluate the mean time with tacrolimus exposure.

3. Transplant Function. The low sample size does not allow to show better renal function in the sirolimus group. Only the Nankivell formula in the ITT and on the therapy population showed the difference. I suggest the conclusion based only the more contemporaneous formula as MDRD. Addictionally, the analysis should be done by mixed models. This type of analysis consider the repeated measures of the data.

The best method is as longitudinal data analysis of covariance with na interaction between time and the treatment variable.

Ref: Different ways to estimate treatment effects in randomised controlled trials. Contemp Clin Trials Commun. 2018 Mar 28;10:80-85.

4. Survival analysis: The plots of graft and patient survival seems to violate the proportional Hazard assumption. The Schoenfeld Residuals Test should be done to test the proportional assumption. A more complex model should be used to incorporate the diferences in event-proportions during the follow-up like a time-dependent proportional Cox Model. The conclusion of better graft survival in the sirolimus group must be done by a Time Dependent Cox Model.

Ref: Time-dependent covariates in the Cox proportional-hazards regression model. Fisher LD. Annu Rev Public Health. 1999;20:145-57.

5. Discussion. The authors should highlight the limitations of the very low rate of patients on the original immunosuppression at the end of 8-9 years follow-up. The conclusions about dnDSA must consider that the majority of the patients were in use of tacrolimus that was not the original immunosuppression.

The conclusion about better renal function in the sirolimus group should be re-analysed based on eGFR by MDRD or CKD-EPI. The conclusion about graft survival probably needs a Cox Model.

Reviewer #2: This is the long-term extension of a previous RCT aimed at evaluating early full conversion from a CNI- to an mTORi-based immunosuppressive protocol as compared with standard CNI-based immunosuppression in KTRs. Main findings of this study are that dnDSA developed similarly in both groups, and long-term renal graft function was better in SRL than in CNI, even though discontinuation rate was significantly higher under mTORi.

The manuscript is of interest for the audience of clinicians involved in the field of kidney transplantation, and findings refer to a very prolonged follow-up. Moreover it is well-written and statistical approach is robust.

However, it lacks originality, since main findings has been previously reported in larger cohorts. Moreover, full conversion from CNI to mTORi appears an old approach, since most studies aimed at addressing the protective role of mTORi on graft function are designed in order to compare mTORi and low exposure CNI vs standard exposure CNI therapy. Last, SRL vs CsA appears also a dated protocol comparison.

Other points:

As correctly acknowledged by the AA a veray graet incidence of drop-out is a limitation (about 50% of the original cohort), since can significantly affects reliability of results.

Is there any relationship between dnDSA development and eGFR changes over time? Indeed AA only present data on the relationship of endpoint eGFR and dnDNA.

Was AR, whose incidence was greater in the CsA group, associated with dnDSA development and/or graft survival?

Any data on proteinuria in the mTORi group?

Reviewer #3: This study reported long term results of the SMART trial (CNI free immunosuppression of SRL, MMF and Pred compared to CsA, MMF and Pred). The results also included development of denovo DSA in addition to GFR, graft and patient survival.

There is significant drop out which is reasonable for this long term observational study, however the reasons for drop out are not addressed or explained. The therapy discontinuation in the Sirolimus group is significantly high compared to the controls (21% vs 66%), which would impact on the final conclusions. The number of patients with donor specific HLA Abs are also small to make a significant conclusion (5 vs 9), however as noted there is no impact type of immunosuppression on dnDSA.

The univariate analysis does not identify any significant risk factor as this study was a long term observation which by default had survived the worse outcomes irrespective of the risks and thus is a selection bias. This aspect or limitation should be explained in the discussion. The benefits of sirolimus in preserving GFR and long term graft survival is demonstrated again in this follow up.

I would recommend addressing these limitations in the discussion and also revising the the language in the Introduction and Discussion sections to conventional manuscript format.

Reviewer #4: This study describes the long-term results (about 9 years) of SMART-Trial, focusing on the development of dnDSA.

Despite the widespread importance of knowing long-term efficacy and safety results of immunosuppressive strategies with mTOR inhibitors, some aspects should be pointed out:

1. The strategy used in SMART-Trial was promising in 2006, but it is currently an exception immunosuppressive regimen: CsA instead of TAC + preemptive conversion rather than mTORi de novo + loading-dose SRL. This makes these results less interesting today.

2. Focusing on dnDSA as the primary objective, trying to correlate the results with the immunosuppressive regimen, does not seem appropriate, since:

a) Only 34 of 140 patients originally enrolled were on therapy.

b) Over such a long follow-up period, many non-evaluated variables may interfere on results, such as: drug conversions, immunosuppression exposure (doses/concentration / variability), and compliance with treatment.

c) There is no information on pre-formed DSA, which makes it difficult to conclude that a first measured DSA is de novo.

I believe a more complete descriptive analysis of efficacy and safety aspects of the 140 patients originally enrolled in SMART trial would be most useful.

Minor issues:

1. I suggest inserting the legends of abbreviations used in the tables and figures

2. I suggest pre-define the abbreviations used throughout the text: Tx, n.s., pts, Abs, SCr-Tk+7, CAN, DCGS, etc

3. I suggest caution in concluding on renal function based on formulas not used today, as Cockroft-Gault and Nankivell.

4. I suggest adjusting the analysis of renal function for deaths, graft losses and follow-up losses.

5. I suggest to pre-define the adverse events listed in Table 7.

6. I suggest that the Discussion session should not be used to repeat results but for critical analysis of study findings.

7. I suggest improving Figure 1 to better demonstrate how many and when patients died, lost their grafts, lost the follow-up and discontinued drugs.

Reviewer #5: The authors updated a German RCT (SMART Trial) with about half (71/140) of the original study group. Blood samples were obtained an average of 8.7 years after initial enrollment in order to screen for the development of DSA. They concluded that “An early conversion to SRL does not result in an increased incidence of dnDSA nor increased long-term risk for the recipient. Transplant function remains improved with benefits for the graft survival.” The important signals reported for the studied survivors were no significant increase in DSA (SRL 5/38, 13.2% vs. CsA 9/33, 27.3%; P=0.097) with an improved eGFR (SRL 64.37 vs. CsA 53.19 ml/min/1.73m2, p=0.044), for those converted to sirolimus. Additional signals that are very encouraging were trends to lower rates of de novo cancers and perhaps fewer viral infections.

In many prior trials using mTORi based immunosuppression to reduce or eliminate CNI drugs very good outcomes were reported for those that tolerated the mTORi. However, prior trials were often plagued by dropouts and investigator switches off the mTORi drugs for various side effects including oral lesions, dyslipidemia, edema, lymphoceles, wound disruptions/hernias, GI intolerance, etc. Many such switches seemed arbitrary and inconsistent, and in effect prevented the formation of firm conclusions from many well-designed trials.

1. While offering some reassuring long-term data that DSA formation is not excessive for those recipients remaining on an mTORi in a CNI free regimen, this sub study only reports 31% of the SRL recipients still taking the drug, and many were on another CNI. So again, why were so many switched? A small table listing the reasons would be informative. Just listing late adverse events is not as informative as listing those that caused a change in drug delivery.

2. No doubt many of the switches off SRL were done due to the side effects caused similarly by mycophenolate mofetil; especially bone marrow suppression and GI intolerance. This should also be noted.

3. Any critical analysis of long-term renal allograft function, especially on an mTORi requires demonstration of the presence and magnitude of proteinuria. I found no such data which is a major limitation of this paper. This should be provided, at least UA dipsticks, or featured as a major limitation.

4. In addition to #3, the absence of histology of the grafts also limits the ability to make firm conclusions on the impact of CNI free status on graft stability, This limitation should be provided as well.

Reviewer #6: This paper looks at a long term follow-up of a clinical trial. Roughly half the patients originally enrolled are available for this follow-up study - the most common reasons for not being in the long-term follow-up study are death and progression.

It is reassuring to note there appears to be no differential dropout by arm; but it would also be useful in Table 1a to look at the comparison of the ITT group and the long-term group, because this will show how the trial population has changed over time.

Please do not use p=n.s. but give actual p-values in the text as they are there in the Tables. The precision should be the same also - 1sf below p=0.01 is fine.

In Table 5-6 there are clear differences between arms in the "on therapy" population - I think these figures should be given in summary only and not tested given the known imbalance (p=0.004) identified earlier - this would invalidate the underlying statistical assumption.

Please give HR & CI for survival to see if this is evidence of lack of effect or lack of evidence of effect. Similarly for graft survival - if p<.05 represents significance then please do not talk of trends to wards or differences when p>.05.

Figure 3 and 4 require total numbers of events per arm to show the power to detect a difference.

6. PLOS authors have the option to publish the peer review history of their article (what does this mean?). If published, this will include your full peer review and any attached files.

Reviewer #1: Yes: Luis Gustavo Modelli de Andrade

Reviewer #2: No

Reviewer #3: No

Reviewer #4: No

Reviewer #5: No

Reviewer #6: No

---

## [Author Response · Author response to Decision Letter 0]

7 Mar 2020

Please find my response to the editor's and reviewers's request uploaded (Andrassy SMART response.doc) along with the revised version of the manuscript

---

## [Decision Letter · Decision Letter 1]

10 Apr 2020

PONE-D-19-25265R1

Early conversion to a CNI-free immunosuppression with SRL after renal Transplantation – long-term follow-up of a multicenter trial

PLOS ONE

Dear Prof. Dr. Andrassy,

Thank you for submitting your manuscript to PLOS ONE. After careful consideration, we feel that it has merit but does not fully meet PLOS ONE’s publication criteria as it currently stands. Therefore, we invite you to submit a revised version of the manuscript that addresses the points raised during the review process.

ACADEMIC EDITOR: 

Please address all issues pointed out by Reviewers 4 and 5. 

Please provide a more detailed analysis of the subgroup of patients maintained on SRL-MMF-Cs for the 8 years

We would appreciate receiving your revised manuscript by May 25 2020 11:59PM. To enhance the reproducibility of your results, we recommend that if applicable you deposit your laboratory protocols in protocols.io, where a protocol can be assigned its own identifier (DOI) such that it can be cited independently in the future. For instructions see: http://journals.plos.org/plosone/s/submission-guidelines#loc-laboratory-protocols

We look forward to receiving your revised manuscript.

Kind regards,

Justyna Gołębiewska

Academic Editor

PLOS ONE

Reviewers' comments:

Reviewer's Responses to Questions

**Comments to the Author**

1. If the authors have adequately addressed your comments raised in a previous round of review and you feel that this manuscript is now acceptable for publication, you may indicate that here to bypass the “Comments to the Author” section, enter your conflict of interest statement in the “Confidential to Editor” section, and submit your "Accept" recommendation.

Reviewer #1: All comments have been addressed

Reviewer #2: All comments have been addressed

Reviewer #3: All comments have been addressed

Reviewer #4: (No Response)

Reviewer #5: (No Response)

Reviewer #6: All comments have been addressed

2. Is the manuscript technically sound, and do the data support the conclusions?

Reviewer #1: Yes

Reviewer #2: Yes

Reviewer #3: Yes

Reviewer #4: Partly

Reviewer #5: Yes

Reviewer #6: (No Response)

3. Has the statistical analysis been performed appropriately and rigorously? 

Reviewer #1: Yes

Reviewer #2: Yes

Reviewer #3: Yes

Reviewer #4: Yes

Reviewer #5: Yes

Reviewer #6: (No Response)

4. Have the authors made all data underlying the findings in their manuscript fully available?

Reviewer #1: Yes

Reviewer #2: Yes

Reviewer #3: Yes

Reviewer #4: Yes

Reviewer #5: No

Reviewer #6: (No Response)

5. Is the manuscript presented in an intelligible fashion and written in standard English?

Reviewer #1: Yes

Reviewer #2: Yes

Reviewer #3: Yes

Reviewer #4: Yes

Reviewer #5: Yes

Reviewer #6: (No Response)

6. Review Comments to the Author

Reviewer #1: I considered that the authors address all the suggestions. The statistic corrections were made. Also, the limitations were discussed in more detail. I considered that the manuscript may be considered for publication in the current revised form.

Reviewer #2: The AA adequately addressed all the concerns raised by this reviewer.

The manuscript was significantly improved

Reviewer #3: The authors have addressed the criticisms about the dropouts in their revision and also explained in the reason for lack of data on proteinuria after 36 months. The manuscript has also been changed to meet the conventional format. The authors have also changed the estimation of GFR to CKD-epi and MDRD. This study has merit in terms of the long follow up, and the limitation would be as mentioned buy the authors the analysis of DSAs was limited to 49% of the study population.

Reviewer #4: Authors have clarified several points raised by reviewers.

They also included important information in the text.

My adicional comments are attached.

Reviewer #5: The authors have provided extension data from a prior RCT termed SMART including 71 of the original ITT 140 kidney transplant recipients with a mean 8.7 year follow-up. The aim of this extension was to detect de novo DSA formation comparing those on a CNI free regimen after a 3-week switch from CsA to Sirolimus to those that remained on CsA; for each MMF/Cs continued. They report that a statistically significant increase in dnDSA was not detected (SRL 5/38, 13.2% vs. CsA 9/33, 27.3%; P=0.097) and GFR remained improved under SRL with (64 vs. 53 ml/min/1.73m2, p=0.044). Although graft failure and skin tumors were less for those on SRL, the difference was not statistically significant. From these data they concluded “An early conversion to SRL does not result in an increased incidence of dnDSA nor increased long-term risk for the recipient. Transplant function remains improved with benefits for the graft survival.”

The study and its results, like many other long-term kidney transplant trials, especially including mTORi suffer from incomplete and perhaps unreliable patient selection and retention. To their credit the authors freely point out that the use of mTORi have been plagued by a lack of tolerability for many patients, and the very inconsistent role more peripheral investigators play in drug switching and dose changes. Their rather sober tally of only “31.6% (12/38) once randomized to receive SRL compared to 66.7% (22/33) started on CsA were still on the original immunosuppressant. The majority had stopped the SRL relatively early as only 40.6% had remained “on therapy” after the first 3 years.” That said mean 8.7 year follow-up data is about the best one could expect to see and MMF switches and changes can be 50% or more as well. More generalized themes are more likely to be true.

Comments

1. The paper often uses 3-month for the time of conversion to SRL—wasn’t it 3 weeks? Fix this in the narrative.

2. Fix your Figure 1 flow sheet and list how many were on sirolimus-MMF-Cs and how many were on CNI-MMF-Cs. How many were truly CNI-free?

3. Those who remain CNI-free on an mTORI usually demonstrate a statistically significant reductions in both skin and other cancers over time, especially after 3 years. The lack of significance in your study, I speculate, is due to the very few that are CNI-free and the larger numbers that switched back to a CNI. Comment.

4. The Discussion should add a comment that dnDSA may have more to do with the stringency of HLA mismatches rather than the choice of maintenance immuno-suppression. The emerging powerful message from eplet matching would seem to support this hypothesis. AJT 2013; 13: 3114–3122 and JASN 2017; 28: 3353–3362.

5. The authors really need to cull out the 40% or so that began and remained on SRL-MMF-Cs for the 8 years. What were their eGFR and urine protein excretion? How many experienced biopsy confirmed rejection and did any of these develop cancer?

Reviewer #6: (No Response)

7. PLOS authors have the option to publish the peer review history of their article (what does this mean?). If published, this will include your full peer review and any attached files.

Reviewer #1: Yes: Luis Gustavo Modelli de Andrade

Reviewer #2: No

Reviewer #3: No

Reviewer #4: No

Reviewer #5: No

Reviewer #6: No

---

## [Author Response · Author response to Decision Letter 1]

4 May 2020

Please find our "Point by Point Response" to your requests under "Andrassy SMART Response II" which is enclosed in this resubmission.

---

## [Decision Letter · Decision Letter 2]

27 May 2020

Early conversion to a CNI-free immunosuppression with SRL after renal Transplantation – long-term follow-up of a multicenter trial

PONE-D-19-25265R2

Dear Dr. Andrassy,

We are pleased to inform you that your manuscript has been judged scientifically suitable for publication and will be formally accepted for publication once it complies with all outstanding technical requirements.

With kind regards,

Justyna Gołębiewska

Academic Editor

PLOS ONE

Additional Editor Comments (optional):

Reviewers' comments:

Reviewer's Responses to Questions

**Comments to the Author**

1. If the authors have adequately addressed your comments raised in a previous round of review and you feel that this manuscript is now acceptable for publication, you may indicate that here to bypass the “Comments to the Author” section, enter your conflict of interest statement in the “Confidential to Editor” section, and submit your "Accept" recommendation.

Reviewer #1: All comments have been addressed

Reviewer #2: All comments have been addressed

Reviewer #3: All comments have been addressed

Reviewer #4: All comments have been addressed

Reviewer #5: All comments have been addressed

Reviewer #6: All comments have been addressed

2. Is the manuscript technically sound, and do the data support the conclusions?

Reviewer #1: Yes

Reviewer #2: Yes

Reviewer #3: Yes

Reviewer #4: Yes

Reviewer #5: Yes

Reviewer #6: (No Response)

3. Has the statistical analysis been performed appropriately and rigorously? 

Reviewer #1: Yes

Reviewer #2: Yes

Reviewer #3: Yes

Reviewer #4: Yes

Reviewer #5: Yes

Reviewer #6: (No Response)

4. Have the authors made all data underlying the findings in their manuscript fully available?

Reviewer #1: Yes

Reviewer #2: Yes

Reviewer #3: Yes

Reviewer #4: Yes

Reviewer #5: Yes

Reviewer #6: (No Response)

5. Is the manuscript presented in an intelligible fashion and written in standard English?

Reviewer #1: Yes

Reviewer #2: Yes

Reviewer #3: Yes

Reviewer #4: Yes

Reviewer #5: Yes

Reviewer #6: (No Response)

6. Review Comments to the Author

Reviewer #1: All comments have been answered by the authors. I considered the manuscript suitable for publication in the present form.

Reviewer #2: The Authors adequately addressed all queries raised by the Reviewers and manuscript is significantly improved

Reviewer #3: The authors have addressed the comments regarding statistical changes and made necessary changes to the manuscript. I am of the opinion that the manuscript be considered for publication in the current format.

Reviewer #4: The authors addressed all questions/concerns raised by the reviewer.

Manuscript sounds suitable to be published in that format.

Reviewer #5: I appreciate the detailed additions made by the authors to update the actual data collected. The long-term followup does add substantial new information to the original study.

Reviewer #6: (No Response)

7. PLOS authors have the option to publish the peer review history of their article (what does this mean?). If published, this will include your full peer review and any attached files.

Reviewer #1: Yes: Luis Gustavo Modelli de Andrade

Reviewer #2: No

Reviewer #3: No

Reviewer #4: No

Reviewer #5: No

Reviewer #6: No

---

## [Editor Report · Acceptance letter]

23 Jul 2020

PONE-D-19-25265R2 

Early conversion to a CNI-free immunosuppression with SRL after renal Transplantation – long-term follow-up of a multicenter trial 

Dear Dr. Andrassy:

I'm pleased to inform you that your manuscript has been deemed suitable for publication in PLOS ONE. Congratulations! Your manuscript is now with our production department. 

Kind regards, 

on behalf of

Dr. Justyna Gołębiewska 

Academic Editor

PLOS ONE